# Insights on Health and Food Applications of *Equus asinus* (Donkey) Milk Bioactive Proteins and Peptides—An Overview

**DOI:** 10.3390/foods9091302

**Published:** 2020-09-15

**Authors:** Reda Derdak, Souraya Sakoui, Oana Lelia Pop, Carmen Ioana Muresan, Dan Cristian Vodnar, Boutaina Addoum, Romana Vulturar, Adina Chis, Ramona Suharoschi, Abdelaziz Soukri, Bouchra El Khalfi

**Affiliations:** 1Laboratory of Physiopathology, Molecular Genetics & Biotechnology, Faculty of Sciences Ain Chock, Health and Biotechnology Research Centre, Hassan II University of Casablanca, Maarif B.P 5366, Casablanca, Morocco; reda.derdakk@gmail.com (R.D.); sakouisouraya@gmail.com (S.S.); boutaina.addoum636@gmail.com (B.A.); a_soukri@hotmail.com (A.S.); bouchra.elkhalfi@gmail.com (B.E.K.); 2Department of Food Science, University of Agricultural Science and Veterinary Medicine, 3-5 Calea Mănăștur, 400372 Cluj-Napoca, Romania; oana.pop@usamvcluj.ro (O.L.P.); carmen.muresan@usamvcluj.ro (C.I.M.); dan.vodnar@usamvcluj.ro (D.C.V.); 3Molecular Nutrition and Proteomics Lab, CDS3, Life Science Institute, University of Agricultural Science and Veterinary Medicine, Calea Mănăștur 3-5, 400372 Cluj-Napoca, Romania; 4Food Biotechnology and Molecular Gastronomy, CDS7, Life Science Institute, University of Agricultural Science and Veterinary Medicine, Calea Mănăștur 3-5, 400372 Cluj-Napoca, Romania; 5Department of Molecular Sciences, ‘Iuliu Hațieganu’ University of Medicine and Pharmacy, Cluj-Napoca, 8 Victor Babeș, 400012 Cluj-Napoca, Romania; adinachis82@gmail.com; 6Cognitive Neuroscience Laboratory, Department of Psychology, Babeș-Bolyai University, Cluj-Napoca, Romania, 1 Mihail Kogalniceanu, 400084 Cluj-Napoca, Romania

**Keywords:** donkey milk (DM), donkey colostrum (DC), mammal’s milk, cow’s milk protein allergy (CMPA), bioactive peptides, biologic activity, immunosenescence, health benefits

## Abstract

Due to its similarity with human milk and its low allergenic properties, donkey milk has long been used as an alternative for infants and patients with cow’s milk protein allergy (CMPA). In addition, this milk is attracting growing interest in human nutrition because of presumed health benefits. It has antioxidant, antimicrobial, antitumoral, antiproliferative and antidiabetic activity. In addition, it stimulates the immune system, regulates the gastrointestinal flora, and prevents inflammatory diseases. Although all donkey milk components can contribute to functional and nutritional effects, it is generally accepted that the whey protein fraction plays a significant role. This review aims to highlight the active proteins and peptides of donkey milk in comparison with other types of milk, emphasizing their properties and their roles in different fields of health and food applications.

## 1. Introduction

Since ancient times, donkey milk (DM) has been known for its therapeutic properties and it was used for wound healing and for treating various diseases such as bronchitis, asthma, joint pain, gastritis [1,2]. Today, it is available on the market as a commercial product to benefit newborns, people with allergies to cow’s milk proteins, and older people [1,2]. Several types of milk (from goat, dromedary, donkey, and horse) are known to have lower allergenicity than cow milk, and it has been suggested that differences in nitrogen distribution and digestibility of milk proteins play an important role in determining the allergenic capacity of milk [3]. DM has become increasingly attractive due to its biological activities, such as anti-microbial, anti-viral, anti-inflammatory, antiproliferative [1,4], and antioxidant activity [5].

The characterization of milk’s main constituents has fundamental importance according to the correlation between health and nutrition. In this context, proteins and peptides are considered important nutrients because some of them show bioactivity when they are native [2,5]. There is increasing evidence that many milk proteins and peptides (particularly peptides that are called ‘*bioactive peptides*’) have physiological functionality. Important effects on immune modulation, cardiovascular health, tumours, bones, and teeth have been frequently reported [5,6]. It is well known that the validity of these effects and the efficacy of functional foods based on bioactive peptides remain to be fully proven in the future, and they may lead to a new class of functional foods, for example, those based on milk proteins and their products [6,7]. 

According to their different solubility, the DM proteins are classified into three classes: milk fat globule membrane (MFGM) proteins, caseins, and whey proteins [4,5]. The protein content of milk may vary among species, among breeds within the same species, and even among individual animals within the same breed. Furthermore, it is well known that there is a strong qualitative resemblance between the principal classes of proteins (i.e., caseins and whey proteins) in all types of milk. These whey proteins and caseins could have biomedical applications [8,9]. Due to its high nutritional and health importance, DM is rediscovered as a functional food. Most of the studies aiming to evaluate DM qualities have been carried out in Italy, and some data are published on the DM obtained from Chinese and Balkan donkey breeds [1]. Likewise, due to the increasingly global spread of food allergies, consumers have started looking for so-called “natural milk” with good taste and useful in treating of some conditions such as cow’s milk protein allergy. This review aims to report published data about the proteins and peptides from DM compared with other kinds of milk (cow, goat, camel, and human milk) and, on the other hand, compared with donkey colostrum. It also shows their biological activities such as anti-microbial, anti-oxidant, anti-inflammatory, anti-allergic, anti-tumoral, anti-obesity, and anti-diabetic activity, their applications in different fields and the benefit of ingesting donkey milk proteins and peptides. 

## 2. Global Composition of Donkey Milk Compared to Other Types of Milk

Due to its chemical composition, milk is considered a complete food. It consists of water, carbohydrates, fats, proteins, and other minor components such as hormones, vitamins, minerals, cytokines [1,5]. Among the constituents of milk, proteins vary between mammalian species, ranging from 1% to 24%. These proteins exist under three categories of proteins defined by their chemical composition and their physical properties: MFGM proteins, caseins, and whey proteins. In addition, the carbohydrate (lactose) milk content varies from 0.7% to 7.0% between different species of mammals. Regarding the fat content, not only the concentration varies but also the chemical composition [4,5]. Table 1 represents the global composition of the donkey, cow, camel, goat, and human milk. The amount of DM components, such as whey protein, lactose, and caseins, are similar to that of human milk, although they differ significantly compared to cow, goat, and camel milk. The only significant difference between DM and human milk is the fat content, which is very low in DM. Nevertheless, regarding the casein-to-whey protein ratio, in DM this is intermediate between human milk and cow milk.

## 3. Bioactive Proteins and Peptides in Donkey Milk Compared to Other Types of Milk 

Table 2 shows the different protein fractions identified in cow, donkey, goat, camel, and human milk, and the g/L amount. Camel’s milk, cow’s milk, and the goat’s type have high protein levels compared to human milk and DM. Cow’s milk, camel’s milk, and goat’s milk have more caseins (80%) and fewer whey proteins (20%) [8] compared with DM, which has more whey proteins (60%) and fewer caseins (40%) [12]. Donkey’s milk has a quantity of α-lactoglobulin resembling that identified in human milk and has a high level of β-lactoglobulin, which is not found in human milk. This β-lactoglobulin is the major allergen of cow’s milk, besides caseins [5,8]. Another peculiarity is that lysozyme’s human and donkey milk content is much higher than in cow milk:

Several research groups have been able to characterize the protein fractions of whey in DM and have demonstrated their nutraceutical properties and their beneficial properties for human health. These proteins will be described in detail below.

### 3.1. Caseins 

The caseins are organized into micelles (supramolecules of colloidal size) whose diameter varies from 30 to 600 nm. In particular, αS1-, αS2-, β-, κ-casein, and traces of γ-casein can be found. These micelles are made up of different proteins (94%) and colloidal calcium phosphate, made of calcium, phosphate, magnesium, and nitrate, comprises 6%. These different caseins have hydrophilic regions and other hydrophobic regions that are different from one casein to another [13]. Besides, the caseins are phosphoproteins. Therefore, they have phosphorylated regions at the level of the serine residues. The proline residues, uniformly distributed into the casein structure, prevent secondary structures such as α helices or β sheets, hence the so-called open or “random coil” conformation of casein [14]. κ-caseins have a particular role, they are first of all glycoproteins and have only one phosphoserine group, but above all, they are stable in the presence of calcium ions and thus protect all of the caseins against precipitation and stabilize the micelles [13,15].

In mature cows’ milk, caseins make up 80% (*w*/*w*) of all proteins, whereas, in humans [16] and equines [17], they represent only 35% and 50% of the total protein content, respectively. The DM essentially comprises αS1-, and β-casein while αS2- and κ-casein are minor components. β-casein can represent up to 80% of the total casein in human milk [18] and is also the predominant protein in the casein fraction of DM [16]. Several studies revealed that caseins and β-lactobglobulin are the main allergens in cow milk [19], and the low allergenicity of DM is explained by low casein content [5].

### 3.2. β-Lactoglobulin

β-lactoglobulin, a globular protein containing 162 amino acids that belongs to the family of lipocalin proteins, has a molecular mass of 18.36 kDa. Lipocalin molecules have pockets capable of hosting iron complexes. Iron binds to protein through iron chelators called “siderophores” [5]. β-lactoglobulin is known for its richness in lysine, leucine, glutamic acid, and aspartic acid. Its secondary structure is mainly composed of β sheets (≈50%), but there are also α helices (10%), β elbows (8%), and a high proportion of disordered structures (35%) [20]. Its structure is also reinforced by two disulfide bridges and a tertiary structure mainly composed of antiparallel β sheets. Studies showed that two different isoforms of β-lactoglobulin could be found in DM: the major isoform is β-lactoglobulin I (80%), while β-lactoglobulin II is encountered in lower quantities. 

In DM, the β-lactoglobulin content of 3.75 g/L resembles that found in cow’s milk [7], and is lower than that found in goat’s milk, while it is absent in camel’s [21] and human’s milk [3]. In DM, β-lactoglobulins correspond to one genetic variant of β-LGI (β-LGIB), two genetic variants of β-LGII (β-LGIIB, and β-LGIIC), and a third minor β-LGII variant (β-LGIID) [5].

β-lactoglobulin is known to have several functions, both nutritional and functional. One of the most studied functions is the protein’s ability to bind specific nutritional interest molecules and serve as a protective matrix during digestion. β-lactoglobulin was shown to bind specific vitamins (D2, D3), cholesterol, particular catechins, and even mercury [20,22]. These interactions occur mainly in the protein’s central area, denominated as the calyx (also known as β-barrel), and formed of β sheets. This hydrophobic cavity, which makes it possible to fix a large variety of ligands, is regulated by an EF loop, which works as a gate to the site of binding. At low pH, this loop is in the “closed” position, and interactions are impossible. When the pH increases, the loop opens, allowing the ligands to insert into the hydrophobic cavity [20]. This change in the Tanford transition structure generally occurs between pH 6.5 and 7.5 [23].

### 3.3. α-Lactalbumin

The α-lactalbumin, a protein composed of 123 amino acid residues, with a molecular weight of 14.2 kDa, has in its tertiary structure four disulfide bridges. Native α-lactalbumin is made up of two distinct domains, and a large section is produced of α helices and a small β sheet domain. A calcium fixation loop bonds the two sections. This protein is found in DM as two isoforms with different isoelectric points (pI) values: 4.76 and 5.26. α-lactalbumin content in DM is 1.8 g/L, a value very close to that found in cow and human milk [4,5].

α-lactalbumin is a protein recognized for its nutritional qualities, mainly for infants’ nutrition. First, α-lactalbumin plays an essential role in milk production in mammals because it binds to the enzyme β-1,4-galactosyltransferase and creates the lactose synthase essential for lactose formation. Another important nutritional element of this protein is its high tryptophan content since it is an essential amino acid. This amino acid has demonstrated positive effects on the development of newborns’ brains and nervous systems and contributes to these systems’ functioning as a direct precursor of serotonin or niacin (also known as vitamin B3). Studies have also shown that regular intake of α-lactalbumin in adult subjects makes it possible to increase the plasma quantities of tryptophan, thus improving certain neurological functions (such as attention, cognitive performance, and morning alertness) [24,25]. This protein also has good digestibility and a low allergenic capacity [26,27].

### 3.4. Lysozyme

Lysozyme, or muramidase, is a globular enzyme consisting of 129 amino acids and a class of hydrolases [28]. The latter consists of two domains: a domain composed essentially of α helices and a β anti-parallel sheet and two α helices. Three disulfide bridges provide the three-dimensional configuration of the molecule: two are found in the α-helix domains, while one is located in the β sheet. Lysozyme can catalyze the hydrolysis of the glycoside 1 → 4 bond of peptidoglycans in the bacterial wall and chitin present in fungi walls [1,5].

Two isoforms of lysozyme, which differ by an oxidized methionine at position 79, were described in DM: lysozyme A with a molecular weight of 14.631 kDa and lysozyme B with a molecular weight of 14.646 kDa [29,30].

Compared to human milk, donkey milk has a higher content of lysozyme (1 g/L), while in goat and cow milk, lysozyme is missing [3,5]. Due to the high amount of lysozyme [3,5] and its thermostability [29,31], the DM is resistant to alteration.

### 3.5. Lactoferrin

Lactoferrin is a glycoprotein that belongs to the transferrin family and has a molecular weight of 80 kDa. Its structure is built by two homologous domains, which bind ferric and carbonate ions. The anti-microbial activity of lactoferrin applies to a wide range of Gram-positive and Gram-negative bacteria. On the one hand, it is partly dependent on its capacity to bind iron, resulting in an environment scarce in iron, which limits the bacterial growth; on the other hand, it depends on its capacity to bind to the lipopolysaccharides of bacterial cell walls via its N-terminus, resulting in the permeabilization of the bacterial cells [5,32]. Likewise, once digested in the stomach, lactoferrin is fragmented into small peptides: lactoferrampin and lactoferricin; the second one has an important action against several bacteria, viruses, fungal pathogens, and protozoa. Moreover, this peptide has other activities, such as inhibition of tumor metastasis in mice [33] and induction of apoptosis in THP-1 human monocytic leukemic cells [34].

### 3.6. Lactoperoxidase

Lactoperoxidase (LPO) is an oxidoreductase enzyme and has a protective function against infections by microorganisms. It is found in low concentrations in fresh DM, as well as in human milk (about 100 times lower than in bovine milk) [1]. LPO can catalyze the oxidation of diverse substrates by using hydrogen peroxide. The oxidation products possess bactericidal activity against bacteria (*Mycoplasmas*), and bacteriostatic effects against *Listeria monocytogenes* [4,5]. It has been shown that immunoglobulins found in the milk exert a synergistic effect on the activity of these non-specific anti-microbial factors with LPO, whose anti-microbial activity against *Streptococcus mutans* increases considerably if the system is incubated with secretory IgA. The anti-microbial activity enhancement seems to be due to binding between LPO and immunoglobulins (IgA) [5,7]. The result of this interaction is a stabilization of the enzymatic activity of lactoperoxidase [5].

## 4. Comparison between Donkey Colostrum and Donkey Milk

Several studies have shown that milk composition is different not only between the species but also between each phase of lactation (colostrum and mature milk) [35,36]. Colostrum is the first form of milk obtained directly after the mammal’s birth until the seventh day. Studies of whey proteins from different mammals (human, bovine, camel) have shown that they generally have differences between colostrum and mature milk. However, changes in the composition of DM in the course of lactation were not sufficiently studied. Recently, the compositions, comparisons, and alterations of the proteome in mammals’ milk at various lactation stages have been studied using advanced proteomics technologies [37,38].

Li et al. [37] were able to identify 300 proteins in DM and mature milk colostrum, including 13 and 12 whey proteins expressed only in donkey colostrum and mature milk, respectively (Table 3). They also showed that in the two types of milk, α-lactalbumin, β-lactoglobulin, lysozyme, and the constant region of the heavy chains of immunoglobulins gamma 1 were the main whey proteins. The same study showed that 18 were expressed differentially between colostrum and mature milk among the proteins identified, of which neural epidermal growth factors like type 2, perilipin, thymosin beta 4, cathepsin B, and transforming factor beta, were induced. Fatty acid-binding proteins had higher levels in mature milk. Simultaneously, tetraspanin, amine oxidase, immunoglobulin gamma 1 heavy chain constant region, apolipoprotein B, prothrombin, major histocompatibility complex (MHC) class I antigen, beta-lactoglobulin II, and alphas 2 casein B were higher in colostrum [37].

Other studies have also shown that there are differences between proteins and between other metabolites’ (including lipids’) compositions, which reveal that the composition of DM changes during lactation [39,40,41].

## 5. Anti-Microbial Activity of Donkey Milk 

For DM, various properties were demonstrated, such as anti-bacterial, anti-viral, and anti-fungal activity. Several studies showed that DM has an anti-bacterial property against a wide range of pathogenic bacteria such as *Escherichia coli*, *Salmonella enteritidis*, *Listeria monocytogène*, *Staphylococcus aureus*, *Bacillus cereus*, *Enterococcus faecalis, Shigella dysenteria,* and against some yeasts [5]. The high content of lysozyme in DM is correlated with high anti-bacterial activity against *Listeria monocytogenes* and *Staphylococcus aureus* [8,42,43,44,45], making DM safer, without food-borne pathogenic bacteria, and with a longer self-life. This anti-bacterial activity is due to its high value of anti-bacterial components [46,47,48], mainly some whey proteins such as lysozyme and lactoferrin [44,45] (Figure 1). Since Gram negative bacteria resist lysozyme due to its lipopolysaccharide membrane, the anti-bacterial activity of DM can be explained by two mechanisms; firstly by the specific structure of lysozyme of DM (similar of equine’s lysozyme), which is able to bind to calcium ions that improve its activity against Gram negative bacteria [42,49,50,51]; secondly by a synergistic activity of lysozyme and lactoferrin, because the latter can bind to membrane proteins of Gram negative bacteria, which disrupt the membrane and open the pores to lysozyme, which destroys the glycosidic linkage (N-acetylglucosamine and N-acetylmuramic acid) of peptidoglycans [32,44,52] (Figure 1). Other studies have shown that the immunoglobulins, IgG, IgA, and IgM [30], also contribute to the inhibition of bacterial growth, acting in synergy with lysozyme [53,54] (Figure 1). Saric et al. (2014) [44] have shown that in addition to the immunoglobulins, some fatty acids such as linoleic acid, lauric acid, and oleic acid, when acting synergetically with lysozyme, show an important anti-bacterial activity against Gram-negative and Gram-positive bacteria. 

In addition to its anti-bacterial activity, DM and its whey proteins were tested for their anti-viral activity. Brumini et al. (2013) [57] have demonstrated that they have the ability to inhibit the replication of *Echovirus type 5*, an enterovirus that affects the human gastrointestinal tract. This activity is due to high molecular weight whey proteins such as lactoferrin, LPO, and immunoglobulins (Figure 1). 

Koutb et al. (2016) shown an anti-microbial activity of DM against two dermatomycotic fungi: *Trichophyton rubrum* and *Trichophyton mentagrophytes,* which are the leading causes of inflammatory tinea corporis [58]. Furthermore, the anti-fungal activity of DM has been tested and found to be effective against fungal strains which are pathogenic for humans (Figure 1). A preliminary study on four samples of DM shown that it inhibits mycotic growth, mainly of *Microsporum canis* and *Trichophyton mentagrophytes,* which are more sensitive than *Microsporum gypseum* to DM [59]. 

It should be mentioned that these anti-microbial factors (lysozyme, LPO, and lactoferrin) are relatively identical in different species (Table 4). Still, their quantity and importance can differ considerably. Indeed, in human milk and DM, lysozyme’s content is substantially higher than that of camel, cow, and goat milk, while the LPO is present in small quantities in DM and human milk, but abundant in cow’s milk. Regarding the lactoferrin, its content is higher in human milk, camel’s, and goat’s milk, respectively [5,8,59].

## 6. Antioxidant Activity of Donkey Milk 

DM is known to have an antioxidant activity, which gives it oxidative stability, providing consumer protection. A study comparing the DM, cow milk and DM powder in terms of antioxidant activity has shown that DM has a higher antioxidant capacity than cow milk. It has a high ability to remove anionic superoxide radicals and to eliminate hydroxyl radicals, which are free radicals generated by body metabolism [60]. Simos et al. [61] were able to determine the antioxidant activity of DM using the method of oxygen radical absorbance capacity, and have shown that the principal contributors of this activity are caseins and the hydrophilic antioxidant compounds, such as uric acid and vitamin C.

## 7. Anti-Inflammatory and Anti-Tumoral Activity of Donkey Milk

Donkey milk is a matrix rich with mediators such as lactoferrin, which has anti-microbial and anti-tumoral activity, interferon γ, which stimulates macrophages, natural killer cells, and cytotoxic T cells [62,63]. DM can induce the release of anti-inflammatory cytokines, retaining a condition of immune homeostasis [62]. Yvon et al. (2018) [64] have demonstrated that the treatment of C57BL/6 mice (Crohn’s disease model) with DM has an anti-inflammatory effect by restoring the levels of anti-microbial peptides such as α-defensin and lysozyme, which help to reduce the imbalance of the microbiota. Moreover, other studies have shown that the lactic flora of DM has an anti-inflammatory effect, for example, by the production of nitric oxide by *Lactobacillus farciminis*. This anti-inflammatory activity can also be due to the synergy between this flora and the anti-microbial peptides [65,66] (Figure 2). Another study has shown that DM and colostrum stimulate the secretion of nitric oxide, a potent vasodilator, and therefore prevent atherosclerosis. They demonstrated that DM stimulates the secretion of immunoglobulins G and interleukins (IL) IL-1β, IL-10, and IL-12. In contrast, colostrum stimulates the secretion of immunoglobulins A. They also showed that the two types of milk stimulate the expression of CD25 and CD69 on human peripheral blood mononuclear cells (Figure 2) and thus, may be useful in the treatment of human immunological diseases [19].

Furthermore, other studies have shown that the administration of human milk or DM improves the liver’s anti-inflammatory state by improving the hepatic mitochondrial functions [80]. Trinchese et al. (2018) [81] have shown that TNF-α and IL-1 decreased, while IL-10 levels increased, in the serum and tissues of rats fed with DM, compared to control rats and rats fed with cow milk. The same team showed that oral supplementation with human milk and DM influences the metabolism of glucose and lipids by modulating pro- and anti-inflammatory serum and tissue mediators.

In addition to its anti-inflammatory activity, DM has other physiological functions such as immunoregulatory and anti-tumor activity [82]. Mao and his collaborators (2009) [77] have shown that many DM fractions can stimulate the production of cytokines IL-2, IFN-γ, IL-6, TNF-α and IL-1β from lymphocytes and macrophages (Figure 2). These cytokines influence anti-proliferation by inducing apoptosis of A549 tumor cells (human lung cancer cells) and the differentiation of these A549 tumor cells into normal cells. They also showed that lysozyme has a strong anti-proliferative effect (Figure 2), and therefore, it could be a promising molecule in treating lung cancer [77]. 

It is well known that milk’s role is critical in developing immunosenescence mitigating strategies because bioactive milk components cannot only directly influence the aging immune physiology, but it can also act as a carrier matrix for a variety of milk products. In this regard, several authors have described the effects of goat and donkey milk consumption on healthy-aged volunteers’ serum cytokine profiles. The authors concluded that DM consumption is useful for increasing the immune response in immunocompromised aged patients [71,72] (Figure 2). Identification of these milk-based products as antiaging agents will promote the concept of “*healthy aging*”.

## 8. Anti-Diabetic and Anti-Obesity Activity of Donkey Milk

In addition to its antioxidant, anti-microbial, anti-inflammatory and anti-tumoral activity, DM also has an anti-diabetic effect (Table 5) [61,76,77].

Type 2 diabetes, also known as non-insulin-dependent diabetes, is a metabolic disease characterized by chronic excess blood sugar (hyperglycemia). The leading causes of type 2 diabetes include obesity, dysfunction of β cells, and insulin resistance by peripheral tissues and cells [84]. Due to its higher whey protein content, DM could help to prevent and treat diabetes by improving glucose metabolism and insulin resistance (Figure 2). Besides the fact that DM and fermented donkey milk has a low caloric intake, the anti-obesity and anti-diabetic activity given by α-lactalbumine should be correlated with triglycerides (TGL) levels—that constitute the largest group of milk lipids. The DM (like the horse milk) has only about 80–85% of total lipids, compared with the cow, sheep, and human milk for which TGL represents 97–98% of total lipids (Figure 2). An outstanding feature regarding the content of DM with effects on metabolic status of the cell is the high content in vitamin B_12_ (cobalamin) of DM: 110 (μg/100 g) compared to other types of milk, i.e., 0.07 in human milk, 0.4 and 0.7 or 0.16 (μg/100 g) in cow, sheep or goat milk, respectively [85]. Vitamin B_12_ is a water-soluble essential micronutrient required by all the body cells, and its deficiency in humans has been implicated in many metabolic processes, such as insulin resistance (besides hematological and neurological disorders). Likewise, Trinchese et al. [80,81] have shown that in animals (rats fed with human milk or DM), improved glucose and lipid metabolism could be identified, with modified mitochondria in adult rat skeletal muscle (compared to untreated control animals). These studies found increased muscle and liver levels of a known regulator of lipid metabolism (OEA: N-oleoylethanolamine), and this could contribute to burning fat and protecting the animals against developing certain obesity-associated metabolic and inflammatory sequelaes (Figure 2). These animals had higher energy expenditures and decreased body lipid accumulation via the mitochondrial uncoupling pathway’s mild augmentation. In addition, several authors speculated that diet-associated changes in microbiota and increased levels of butyrate (a short chain fatty acid) in human and DM-fed rats (compared with cow milk fed rats) contributed to the differences in metabolism and mitochondrial function through several as yet unknown signalling pathways [80,81,86]. 

Li et al., in 2020 [76], have shown that DM improves the viability of damaged pancreatic β cells, but does not stimulate the secretion of insulin by damaged β cells and that the α-lactalbumin increases the insulin sensitivity of the target organs. DM has shown a better effect than metformin, an anti-diabetic drug for treating type 2 diabetes [76]. 

Besides, they showed that DM decreased the level of glycosylated hemoglobin and it acted positively in the treatment of diabetes by inhibiting the expression of phosphoenolpyruvate carboxykinase 1 and glucose-6-phosphatase, which are key enzymes in hepatic gluconeogenesis (Table 5).

## 9. Food and Other Applications of Donkey Milk 

Considering its functional properties and its nutritional values, DM becomes an attractive product for health, technology, cosmetic industry, and others. The health effects of DM consumption are mainly related to low allergenicity [3,5,67,68], anti-microbial activity [5,32,42,43,44,57,69,70], regulation of iron homeostasis [5], anti-inflammatory and immune system modulation, innate immune system, immunosenescence [5,20,71,72], anti-hypertensive [5,73,74,75], anti-diabetic [76], anti-tumoral [5,77,78], stimulate development [5,68], anti-stress and anti-oxidant activity [5,25,65,76], and anti-osteoporosis [5,68,79] (Figure 2). DM is supplied in different forms: liquid milk, fermented products (with higher peptide content and bioavailable calcium source), freeze dried, and spray dried powders [87]. The application of new technologies such as freeze drying and microencapsulation allows better exploitation of this product [32]. Several studies have revealed that DM is an adequate alternative to children suffering from cow milk proteins allergy (CMPA) [67,88], due to its low composition of caseins, which constitute the main allergenic components of milk. Sarti et al. (2019) [67] have shown that DM has no negative influence on infants and children, and have assessed its ability to manage the “Food Protein Induced Enterocolitis Syndrome” (FPIES) caused by cow’s milk. However, cow’s milk proteins cause immune reactions mediated by immunoglobulins E (IgE), which is also managed with DM [67,88]. Donkey milk is also an important vitamin D source, which can be obtained only from diet or from exposure to sun-light. The oral source of vitamin D can be important in winter and for people who cannot be exposed to sun-light [88].

Donkey milk is a basic ingredient for the production of high-value dairy products [89]. DM lysozyme is also used in the food industry due to its stability and resistance to various technological processes such as thermic treatment [31] and digestive tract conditions such as acid pH and gastrointestinal enzymes [90].

To value the donkey, Cappola et al. (2002) [91] have investigated DM’s ability for fermentation by *Lactobacillus rhamnosus*, and showed that DM is a good base for probiotics and therapeutic food formulation. In this context, another team used *Lactobacillus casei* in addition to *Lactobacillus rhamnosus* as probiotics to produce a fermented drink made from DM, and these probiotics were able to survive in this milk up to 30 days [92]. Indeed, DM is also a base for producing yogurt products (Standard yogurt and probiotic yogurt), and yogurt supplied with probiotics showed a high antioxidant activity and a low content in lactose, which is beneficial for consumers with an allergy to cow milk proteins [93]. 

Because of its low fat and casein content, DM constitutes a soft gel during cheese manufacturing. Several studies have faced this situation, for example, by the addition of MTGase—a microbial transglutaminase that can enhance the texture of the curd without any effect on moisture, proteins, fats, or cheese yield [89]. Šarić et al. (2016) overcame this limitation by mixing DM with goat milk to produce a functional product with high quality [94].

Since recently, as Salimei 2016 [95] points out, donkey’s milk could thus be placed among the new generation of fermented milk drinks, such as koumiss derived from mare’s milk, and would allow for an effective combination of the advantageous properties of the raw ingredients with lactic acid bacteria of probiotic interest. In these years, other products as ice cream, biscuits, cakes, desserts, and liqueurs have been developed from pasteurized donkey’s milk, and its technological use (2%) has been successfully tested in hard cheese making, contributing to innovation in the dairy sector. Regarding the cosmetic industry, besides its potential roles in human nutrition, multipurpose applications of donkey’s milk are reported in ethnomedicine, and it is used in cosmetology, most likely due to its lysozyme content, effective in smoothing skin and scalp inflammations [95].

## 10. Conclusions

Starting with fundamental importance for the correlation between health and nutrition, this review outlines the importance of the protein fraction of DM. This type of milk was used since ancient Egyptian, Greek, and Roman times, not only for its nutritional value for infants but also for its beneficial skincare properties. Later, it was recognized as a common remedy for many ailments, and in French orphanages during the late XIXth or early XXth century when infants receiving DM grew well and with lower mortality than those given cow milk, as reviewed by Carminati et al. (2017) [28], and Fantuz et al. (2016) [3]. Nowadays, DM is considered a medicinal food (or “*pharma food*“) because of its nutritional and functional properties, and because of having a composition similar to human milk when compared to other types of milk; is known that donkey’s milk has a casein-to-whey protein ratio intermediate between human milk and cow milk. Donkey milk has various biological activities such as vasodilation (through the secretion of nitric oxide and therefore preventing atherosclerosis), stimulation of the immune system, and anti-diabetic, anti-inflammatory, anti-allergic, anti-obesity, anti-proliferative and anti-microbial activities. These activities are specifically attributed to whey proteins such as lactoferrin, LPO, lysozyme, and immunoglobulins. As Carminati et al. (2017) [28] mentioned, regarding the consumption of fermented DM products (with higher peptide content) by the elderly, this should be encouraged due to this excellent source of bioavailable calcium, low caloric intake, and the ability to modulate the aged immune system, including the intestinal mucosal immune response. Besides, applying certain new technologies, such as lyophilization and microencapsulation, allows better exploitation of this animal product. As a recent editorial outlined [49], although there has been a rise in consumer interest about DM (and other dairy products), it is still a ‘niche product’. The food industry should try to increase the production and availability of DM on the market; this will raise awareness and promote this sector, highlighting the food applications with multiple health benefits not only for the growing infants, but for all ages, convalescents and the elderly. For all these benefits and versatility, future production systems have to be committed to combine profitability with responsibility in protecting animal and human welfare and preventing chronic diseases (non-communicable diseases—NCDs). 

## Figures and Tables

**Figure 1 foods-09-01302-f001:**
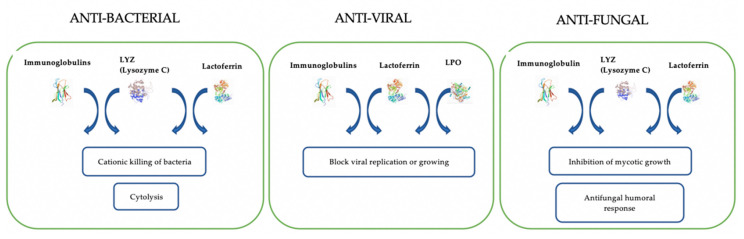
Molecular mechanism of anti-microbial activity of donkey milk (DM) proteins (protein structure is a reference to UniProtKB [55] and the Protein Data Bank—PDB [56] (LYS: Lysozyme C: (P11375); Lactoferrin (A0A3Q9HG40); Immunoglobulin (Q861S3); LPO—Lactoperoxidase (P80025))). The anti-bacterial activity of DM is mainly due to lysozyme, lactoferrin, and immunoglobulins (IgG, IgA, IgM) through two molecular mechanisms: cytolysis and the cationic killing of bacteria (LYS could act synergistically with lactoferrin, and immunoglobulins). The anti-viral capability of DM was related to synergetic action between LYS, LPO, immunoglobulins, and low abundant–low molecular weight protein fraction (<30,000 Da) through blocking viral replication or growing by binding to host cells and/or direct interaction with the viruses. The anti-fungal activity is due to the antifungal capacity of DM’s proteins (lactoferrin, LYS, and immunoglobulins) within the inhibition of mycotic growth and protective cell reactions triggered in response to the fungi presence; (the arrows represent the synergistic activity of proteins).

**Figure 2 foods-09-01302-f002:**
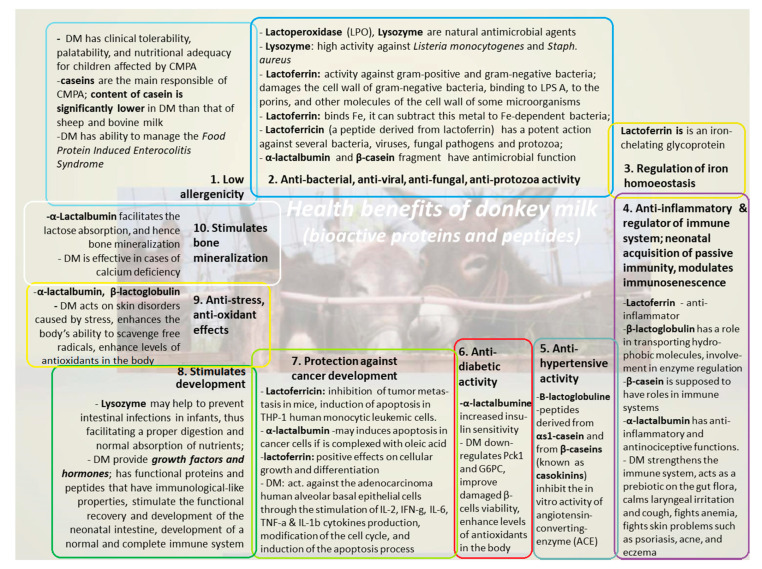
Health implications of donkey milk consumption due to bioactive proteins and peptides; the main ten health properties are listed, outlining the references and mechanisms of bioactive proteins/peptides involved in: **1.** Low allergenicity [3,5,67,68]; **2.** Anti-bacterial, anti-viral, anti-fungal, anti-protozoa activity [5,32,42,43,44,57,69,70]; **3.** Regulation of iron homoeostasis [5]; **4.** Anti-inflammatory and regulator of immune system, neonatal acquisition of passive immunity, immunosenescence [5,20,71,72]; **5.** Anti-hypertensive activity [5,73,74,75]; **6.** Anti-diabetic activity [76]; **7.** Protection against cancer development [5,77,78]; **8.** Stimulate development [5,68]; **9.** Anti-stress, anti-oxidant effects [5,25,65,76]; **10.** Stimulate bone mineralization [5,68,79]. Abbreviations: CMPA—cow milk protein allergy, LPS A—lipopolysaccharide A, Pck1—phosphoenolpyruvate carboxykinase 1, G6PC—glucose-6-phosphatase.

**Table 1 foods-09-01302-t001:** Milk composition and energy value—donkey and other species [5,10,11].

Milk Characteristics	Donkey (%)	Goat (%)	Cow (%)	Camel (%)	Human (%)
Proteins	1.74	3.41	3.43	1.80	1.64
Fat	1.21	4.62	3.46	1.80	3.38
Lactose	6.23	4.47	4.71	2.91	6.69
Dry Matter	9.61	13.23	12.38	11.30	12.43
Ashes	0.43	0.73	0.78	0.85	0.22
Water	90.39	86.77	87.62	90.60	87.57
Energy (KJ/Kg)	1939.40	3399.50	2983.00	2745.80	2855.60

**Table 2 foods-09-01302-t002:** Main proteins of donkey milk compared to other types of milk [5,8,12].

	Cow (g/L)	Donkey (g/L)	Goat (g/L)	Camel (g/L)	Human (g/L)
Total protein content	31–38	13–28	25–39	25–45	9–17
Total casein	27.2	6.6	25	26.4	5.6
Total whey protein	4.5	7.5	6	6.6	8
αS1-casein	10–15	0.2–1	0–7	5	0.3–0.8
αS2-casein	3–4	0.2	4.2	2.2	n.d.
β-casein	9–11	3.9	11–18	12.8	1.8–4
κ-casein	3–4	n.d.	4–4.6	0.8	0.6–1
α-lactalbumin	1–1.5	1.8–3	1.2	3.5	1.9–2.6
β-lactoglobulin	3.3–4	3.2–3.7	2.1	n.d.	n.d.
Lysozyme	0.00007	1	Trace	0.00015	0.04–0.2
Lactoferrin	0.1	0.08	0.02–0.2	0.22	1.7–2
Immunoglobulins	1	n.d.	1	1.54	1.1
Albumin	0.4	n.d.	0.5	0.4	0.4

n.d.—not detected.

**Table 3 foods-09-01302-t003:** Uniquely expressed proteins in donkey colostrum and donkey mature milk [37,39,40,41].

Types of Milk	
Donkey Colostrum	Donkey’s Mature Milk
Zinc-alpha-2-glycoprotein	Histone H3
Immmunoglobulin lambda light chain variable region	Myristoylateda lanine-rich C-kinase substrate
Thrombospondin 4	Histone H4
Peptidoglycan recognition protein 1	Multiple coagulation factor deficiency 2
Cartilage acidic protein 1	C-C motif chemokine
Peptidyl-prolyl cis-trans isomerase	Transcription factor adipocyte enhancer binding protein 1 (AEBP1)
L receptor related protein 1	Follistatin-like 1
Insulin like growth factor binding protein 7	ST6 beta-galactoside alpha-2,6-sialyltransferase 1
Major histocompatibility complex (MHC) class II associated invariant chain	Uncharacterized protein
DNA J-like protein subfamily B member 11-like protein	Multiple coagulation factor deficiency protein-like protein
Cathepsin Z	Uncharacterized protein
Uncharacterized protein	Glutathione peroxidase (Fragment)
Amino peptidase	

**Table 4 foods-09-01302-t004:** Quantity of the major anti-microbial proteins found in human, bovine, camel and donkey milk [5,8,59].

Milk Type	Lysozyme (g/L)	Lactoperoxidase (g/L)	Lactoferrin (g/L)
Donkey	1.0	0.11	0.08
Human	0.12	0.77	0.3–4.2
Goat	Trace	Trace	0.02–2
Camel	0.00015	n.d.	0.22
Cow	0.00007	30–100	0.10

n.d.—not detected.

**Table 5 foods-09-01302-t005:** Biological activity of donkey milk proteins/peptides.

Proteins/Peptide	Biological Activity	Reference
αS1- and β-casein	Antihypertensive	[73]
	Inhibitor activity of Angiotensin converting enzyme (ACE) and antioxidant activity.	[5]
αS1- casein	Transport calcium phosphate	UniProtKB—P86272
αS2- casein	Transport calcium phosphate	UniProtKB—B7VGF9
Lysozyme	Anti-inflammatory properties	[63]
	Anti-tumoral properties	[77]
	Reduced incidence of diarrhea	[83]
	Anti-microbial properties	[32]
Lactoferrin	Antmicrobial properties	[57]
	Antitumor	[73]
	Antithrombotic	[74]
Lactoperoxidase	Antioxydant properties	[61]
	Anti-microbial properties	[57]
α-lactalbumine	Anti-diabetic properties	[76]
	Antihypertensive	[74]
β-lactoglobuline	Antihypertensive	[74]

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
