# Peer review of "Insights on Health and Food Applications of Equus asinus (Donkey) Milk Bioactive Proteins and Peptides—An Overview"

_foods, 2020, doi:10.3390/foods9091302_

Round 1

Reviewer 1 Report

Manuscript entitled „Insights on health and food applications of Equus asinus (donkey) milk bioactive proteins and peptides – An overview” compares  the active peptides and proteins of donkey milk with other types of milk, emphasizing their properties and their roles in different fields of health and food applications.

The manuscript consists of 5 tables, 2 figures, and 88 references.

The tables contain the most important information on the composition of the milk of individual species of animals, and figure 2 presents cumulatively health implications of donkey milk consumption. A very well written and interesting manuscript, systematizes knowledge about different animal species compared to donkey’s milk.

I have only one question: have you ever found any information about A1/A2 beta-casein in donkey’s milk?

Author Response

Reviewer #1: Manuscript entitled „Insights on health and food applications of Equus asinus (donkey) milk bioactive proteins and peptides – An overview” compares the active peptides and proteins of donkey milk with other types of milk, emphasizing their properties and their roles in different fields of health and food applications.

The manuscript consists of 5 tables, 2 figures, and 88 references.

The tables contain the most important information on the composition of the milk of individual species of animals, and figure 2 presents cumulatively health implications of donkey milk consumption. A very well written and interesting manuscript systematizes knowledge about different animal species compared to donkey’s milk. 

I have only one question: have you ever found any information about A1/A2 beta-casein in donkey’s milk?

Response: To answer your question, we have found a paper that refers to A1/A2 β-Casein polymorphisms, that compare β-Casein from different species (including donkey) through multiple alignments. The original β-Casein variant carrying Pro67. [Caroli AM, Savino S, Bulgari O, Monti E. Detecting β-Casein Variation in Bovine Milk. Molecules. 2016;21(2):141. Published 2016 Jan 25. doi:10.3390/molecules21020141]. We do not come to a cross-reference related with donkey milk A1/A2 β-Casein polymorphisms.

Due to adding new citations in the text, the total number of the references is 95.

Reviewer 2 Report

A manuscript entitled “Insights on health and food applications of Equus acinus (donkey) milk bioactive proteins and peptides - An over view” summarize recent scientific advances on donkey’s milk in terms of the nutritional and health beneficial aspects. It seems to be useful for the readers of the Journal “Foods”, but several corrections should be needed as pointed out below.

  • P1L44, References should be added.
  • P2L55, Terminology is confused. It is better to avoid deal peptides as equally to proteins.
  • P2L57, References should be added.
  • P2L62, “proteins” should be added following to “milk fat globule membrane”.
  • P2L62, “The protein content…types of milk” seems to be confused. Are the milk proteins are similar or different among mammal species? 
  • P2L66, “These whey proteins…biomedical applications” does not make sense. Biopolymers are polymers derived from living organisms and they are not necessarily biologically active agents with biomedical applications (some of them are even harmful for human health, such as “dental biofilms”).
  • P2L83, “These proteins…whey proteins” seems to be contradiction to P2L61. It should be better to categorize milk proteins into three as like as the description in P2L61.
  • P4L131, “Iron binds…”siderohpores” gives abrupt impression. Does this “protein” mean beta-lactoglobulin?
  • P4L139, beta-lactoglobulin is absent in camel’s milk as far as I know. See the reference: "Hinz K., O’Connor P.M., Huppertz T., Ross R.P., Kelly A.L. Comparison of the principal proteins in bovine, caprine, buffalo, equine and camel milk. J. Dairy Res. 2012;79:185–191. doi: 10.1017/S0022029912000015”
  • P5L182, “This high…long time” seems too speculative. This is because natural lysozyme or lysozyme alone can exhibit antibacterial activity just towards Gram-positive bacteria as the authors pointed out. Also check P7L238.
  • P5L206. Is this citation correct?
  • P7L247, Which subtype are the “immunoglobulins”
  • P7L250, 251, “gram” should be “Gram”
  • P7L256, “fungus” should be “fungi”
  • P8L290, “lactobacillus” should be “Lactobacillus”
  • P8L300, “2018”should be in parentheses.
  • P9L336, What does “colab.” mean?
  • P11L406, Either “main” or “mainly” should be removed to avoid redundancy. “The main health…anti-osteoporosis (Figure 2).” seems not to be in an appropriate position. It may be better relocate this sentence in the earlier position of section nine. For example, it could be integrated in the first paragraph.
  • Table 1, Values should be aligned by decimal points.
  • Table 2, ?S2-casein and ?-casein should be used.
  • Table 3, Center alignment seems to be inappropriate. It makes different to distinguish each items at a a glance (e.g., substrate, variable region, etc). Furthermore, “uncharacterized protein” appeared in this table indicate the same protein? A line under Glutathione peroxidase (Fragment) in the right column should be removed. “Mature milk” should be used instead of “milk”, and it is the same for entire through the text. 
  • Table 5, ?S1- and ?S2- should be applied.
  • Fig. 1, Hyphenation should be added after “ANTI”. Furthermore, PDB accession numbers should be provided. It is difficult to realize what does the arrow appeared in the leftmost area of the figure mean. Probably some explanation should be needed. All the pilcrows should be removed.

Author Response

Reviewer #2: According to your requirements, we make several corrections that were pointed out, as follows.

Comment 1: P1L44, References should be added.

Response: P1L44, The appropriate citation was added [1, 5]

Comment 2: P2L55, Terminology is confused. It is better to avoid deal peptides as equally to proteins.

Response: P2L64-67, We rephrase to clearly state proteins and peptides, as: “In this context, proteins and peptides are considered important nutrients because some of them show bioactivity when they are native [2, 5]. There is increasing evidence that many milk proteins and peptides (mainly particularly peptides that are called ‘bioactive peptides’) have physiological functionality.”

Comment 3: P2L57, References should be added.

Response: P2L67-68, The appropriate citation was added as follows: “Important effects on immune modulation, cardiovascular health, tumours, health of bones and teeth, have been frequently reported [5, 6].”

Comment 4: P2L62, “proteins” should be added following to “milk fat globule membrane”.

Response: P2L72-73, “proteins” were added as recommended: “milk fat globule membrane (MFGM) proteins”

Comment 5: P2L62, “The protein content…types of milk” seems to be confused. Are the milk proteins are similar or different among mammal species? 

Response: P2L73-75, Milk proteins are different quantitatively but qualitatively there are some similarities between some proteins of different mammal species, such as caseins and whey proteins. We rephrase as follows: “The protein content of milk may vary among species, among breeds within the same species, and even among individual animals within the same breed.”

Comment 6: P2L66, “These whey proteins…biomedical applications” does not make sense. Biopolymers are polymers derived from living organisms and they are not necessarily biologically active agents with biomedical applications (some of them are even harmful for human health, such as “dental biofilms”).

Response: P2L76-77: We rephrase as: “These whey proteins and caseins could have biomedical applications [7, 8].”

Comment 7: P2L83, “These proteins…whey proteins” seems to be contradiction to P2L61. It should be better to categorize milk proteins into three as like as the description in P2L61.

Response: P2L92-93: We correct and categorize milk proteins into three groups as follows: “These proteins exist under three categories of proteins which are defined by their chemical composition and their physical properties: MFGM proteins, caseins, and whey proteins.”

Comment 8: P4L131, “Iron binds…”siderohpores” gives abrupt impression. Does this “protein” mean beta-lactoglobulin?

Response: P4L167-168, Yes, this “protein” means β-lactoglobulin. We correct the initial text as follows: “β-lactoglobulin is known for its richness in lysine, leucine, glutamic acid, and aspartic acid.”

Comment 9: P4L139, beta-lactoglobulin is absent in camel’s milk as far as I know. See the reference: "Hinz K., O’Connor P.M., Huppertz T., Ross R.P., Kelly A.L. Comparison of the principal proteins in bovine, caprine, buffalo, equine and camel milk. J. Dairy Res. 2012;79:185–191. doi: 10.1017/S0022029912000015”

Response: P4L175-176, Yes, beta-lactoglobulin is absent in DM; we add an appropriate citation and make a correction: “In DM, the β-lactoglobulin content is 3.75 g/L resembles that found in cow’s milk, and is lower than that found in goat’s milk, while it is absent in camel’s [20] and human’s milk [3].”

Comment 10: P5L182, “This high…long time” seems too speculative. This is because natural lysozyme or lysozyme alone can exhibit antibacterial activity just towards Gram-positive bacteria as the authors pointed out. Also check P7L238.

Response: P5L231-232: We rephrase the statement as follows: ” Due to the high amount of lysozyme [3, 5] and its thermostability [28] the DM is resistant to alteration.” The appropriate citation was added: [28] Ozturkoglu-Budak S (2016) Effect of different treatments on the stability of lysozyme, lactoferrin, and b-lactoglobulin in donkey’s milk. Int J Dairy Technol 71: 36-45.

We check P6L292: Natural lysozyme or lysozyme alone can exhibit antibacterial activity just towards Gram-positive bacteria as we pointed out. But it has anti-bacterial activity against some Gram negative strain as well, and it can be explained by two mechanisms; firstly by the specific structure of lysozyme of DM (similar of equine’s lysozyme), which is able to bind to calcium ions which improve its activity against Gram negative bacteria, secondly by the synergistic activity of lysozyme and lactoferrin.

Comment 11: P5L206. Is this citation correct?

Response: P5L242-244: The correct citations were added: “Moreover, this peptide has other activities, such as inhibition of tumor metastasis in mice [30] and induction of apoptosis in THP-1 human monocytic leukemic cells [31].” [30] Yoo YC, Watanabe S, Watanabe R, Hata K, Shimazaki KI, Azuma I. Bovine lactoferrin and lactoferricin, a peptide derived from bovine lactoferrin, inhibit tumor metastasis in mice. Japanese Journal of Cancer Research. 1997;88(2):184–190. DOI: 10.1007/978-1-4757-9068-9_35; and [31] Yoo YC, Watanabe R, Koike Y, et al. Apoptosis in human leukemic cells induced by lactoferricin, a bovine milk protein-devived peptide: involvement of reactive oxygen species. Biochemical and Biophysical Research Communications. 1997;237(3):624–628. DOI: 10.1006/bbrc.1997.7199

Comment 12: P7L247, Which subtype are the “immunoglobulins”

Response: P7L300-301, We specify the subtype of immunoglobulins and add citation: “Other studies have shown that the immunoglobulins, IgG, IgA, and IgM [29], also contribute to the inhibition of bacterial growth, acting in synergy with lysozyme [50, 51] (Figure 1). “

Comment 13: P7L250, 251, “gram” should be “Gram”

Response: P7L303-304, We change to Gram as follows: “…in synergy with lysozyme, show an important antibacterial activity against Gram-negative and Gram-positive bacteria.”

Comment 14: P7L256, “fungus” should be “fungi”

Response: P7L311, we change to “fungi” as follows: “Another study showed an antiviral activity of DM against 2 dermatomycotic fungi:…”

Comment 15: P8L290, “lactobacillus” should be “Lactobacillus”

Response: P8L366, we change to “Lactobacillus” as follows: “….., for example by the production of nitric oxide by Lactobacillus farciminis. “

Comment 16: P8L300, “2018” should be in parentheses.

Response: P8L376, we add parentheses as follows: “Trinchese et al. (2018)…”

Comment 17: P9L336, What does “colab.” mean?

Response: P9L413: we change as follows: “Likewise, Trinchese et al. [64, 65] have shown……”

Comment 18: P11L406, Either “main” or “mainly” should be removed to avoid redundancy. “The main health…anti-osteoporosis (Figure 2).” seems not to be in an appropriate position. It may be better relocate this sentence in the earlier position of section nine. For example, it could be integrated in the first paragraph.

Response: P10L441-446, we rephrase and relocate the sentence in the first paragraph of section nine: “The health effects of DM consumption are mainly related to low allergenicity [3, 5, 80, 90], antimicrobial activity [5, 31, 41-43, 54, 91, 92], regulation of iron homeostasis [5], anti-inflammatory and immune system modulation, innate immune system, immunosenescence [5, 19, 70, 71], anti-hypertensive [5, 76, 78, 93], anti-diabetic [72], anti-tumoral [5, 69, 95], stimulate development [5, 90], anti-stress and anti-oxidant activity [5, 24, 64, 72], anti-osteoporosis [5, 90, 94] (Figure 2).”

Comment 19: Table 1, Values should be aligned by decimal points.

Response: Table 1, we aligned the values by decimal points.

Comment 20: Table 2, ?S2-casein and ?-casein should be used.

Response: Table 2, we correct ?S2-casein and ?-casein.

Comment 21: Table 3, Center alignment seems to be inappropriate. It makes different to distinguish each items at a a glance (e.g., substrate, variable region, etc). Furthermore, “uncharacterized protein” appeared in this table indicate the same protein? A line under Glutathione peroxidase (Fragment) in the right column should be removed. “Mature milk” should be used instead of “milk”, and it is the same for entire through the text. 

Response: Table 3, we aligned left the text in the table; the “uncharacterized protein” is different. We removed the line under Glutathione peroxidase (Fragment). We add “mature milk” instead of “milk” in the table (title, and column 2 name), and as well in the entire through the text that compares colostrum and mature milk.

Comment 22: Table 5, ?S1- and ?S2- should be applied.

Response: Table 5, we correct ?S1- and ?S2-

Comment 23: Fig. 1, Hyphenation should be added after “ANTI”. Furthermore, PDB accession numbers should be provided. It is difficult to realize what does the arrow appeared in the leftmost area of the figure mean. Probably some explanation should be needed. All the pilcrows should be removed.

Response: Fig. 1, L361-372, We removed the pilcrows and insert “-“ after ANTI as follows: “ANTI-BACTERIAL”, “ANTI-VIRAL”, and “ANTI-FUNGAL”

The following explanation was added: “Figure 1. Molecular mechanism of anti-microbial activity of DM proteins (protein structure is a reference to UniProtKB [54] and Protein Data Bank – PDB [55] (LYS: Lysozyme C: (P11375); Lactoferrin (A0A3Q9HG40); Immunoglobulin (Q861S3); LPO – Lactoperoxidase (P80025))). The anti-bacterial activity of DM is mainly due to lysozyme, lactoferrin, and immunoglobulins (IgG, IgA, IgM) through two molecular mechanisms cytolysis and cationic killing of bacteria (LYS could act synergistic with lactoferrin, and immunoglobulins).  The anti-viral capability of DM was related to synergetic action between LYS, LPO, immunoglobulins, and low abundant – low molecular weight protein fraction (<30000 Da) through blocking viral replication or growing by binding to host cells and/or direct interaction with the viruses. The anti-fungal activity is due to the antifungal capacity of DM’s proteins (lactoferrin, LYS, and immunoglobulins) within the inhibition of mycotic growth and protective cell reactions triggered in response to the fungi presence; (the arrows represent the synergistic activity of proteins).”

Fig. 2. In order to avoid redundancy, we removed from figure 2 the references (names and years) that are presented into the legend with numbers and into the text in a paragraph above (section 9, P10L441-446).

Due to adding new citations in the text, the total number of the references is 95.
